# Use of Underwater Acoustics in Marine Conservation and Policy: Previous Advances, Current Status, and Future Needs

**Shane Guan** [1,*] **, Tiffini Brookens** [2,*] **and Joseph Vignola** [1]

1. Department of Mechanical Engineering, The Catholic University of America, Washington, DC 20064, USA; vignola@cua.edu
2. U.S. Marine Mammal Commission, Bethesda, MD 20814, USA
* Correspondence: guan@cua.edu (S.G.); tbrookens@mmc.gov (T.B.)

**Abstract:** The interdisciplinary field of assessing the impacts of sound on marine life has benefited largely from the advancement of underwater acoustics that occurred after World War II. Acoustic parameters widely used in underwater acoustics were redefined to quantify sound levels relevant to animal audiometric variables, both at the source and receiver. The fundamental approach for assessing the impacts of sound uses a source-pathway-receiver model based on the one-way sonar equation, and most numerical sound propagation models can be used to predict received levels at marine animals that are potentially exposed. However, significant information gaps still exist in terms of sound source characterization and propagation that are strongly coupled with the type and layering of the underlying substrate(s). Additional challenges include the lack of easy-to-use propagation models and animal-specific statistical detection models, as well as a lack of adequate training of regulatory entities in underwater acoustics.

**Keywords:** underwater acoustics; underwater sound impacts; marine conservation; impact assessment

## 1. Introduction—Historical Perspective

As a visually oriented species, it is not surprising that our knowledge of the world has been based largely on reasoning and experimentation through visual means. This includes research on marine organisms whose natural habitats beneath the ocean surface are mostly beyond visual observation, e.g., [1,2]. Moreover, the lack of an auditory system that functions efficiently underwater led humankind to consider the marine environment as a "silent world" for eons, e.g., [3].

The need for navigational safety, especially after the sinking of the HMS Titanic as a result of striking an iceberg in the North Atlantic Ocean, and to detect enemy submarines (i.e., antisubmarine warfare or ASW) and warships during the two world wars prompted tremendous advancements in the field of underwater acoustics between the 1910s and 1950s [4,5]. However, it was not until the discovery of underwater sound production and orientation [6–8] and communication [9,10] in several cetacean species that marine biologists began to investigate the potential effects of underwater sound on marine mammals.

While it was widely recognized before the 1970s that high sound levels, either from elevated ambient sound or from sonar ping reverberation, could adversely affect signal detection in the naval sonar community [4,5], Payne and Webb [11] were the first to document that long-distance acoustic communication ranges could be greatly reduced as a result of increased background sound levels from ships. By using simple sonar equations, Payne and Webb [11] determined that, with the advent of propeller-driven ships, the transmission range of 20-Hz fin whale calls was reduced by nearly 100 and 3000 nmi using spherical and cylindrical spreading models, respectively.

Some of the greatest achievements in environmental conservation in the United States occurred in the 1970s with the enactment of various laws, including passage of

the Marine Mammal Protection Act (MMPA) in 1972 and the Endangered Species Act (ESA) in 1973 [12–14]. Implementation of the MMPA led to strict regulations by U.S. Federal agencies to reduce the incidental taking of marine mammals, initially in regard to fishery bycatches [15,16].

The need to implement measures to mitigate impacts on marine mammals from human activities beyond commercial fisheries led to many government-sponsored studies and workshops. For example, in the early 1980s through the 1990s, the U.S. Minerals Management Service (MMS, the predecessor of the current Bureau of Ocean Energy Management (BOEM)) funded several studies investigating disturbance of marine mammals from oil and gas development activities in the Arctic [17]. Many of those studies provided novel information on how underwater sound from various industrial activities affected marine mammals [18–21].

In the early to mid-1990s, two global oceanographic experiments became controversial based on the concern that the intense underwater sound used for climate research would harm whales [22]. The Heard Island Feasibility Test (HIFT) and the subsequent Acoustic Thermometry of Ocean Climate (ATOC) used sufficiently intense low-frequency signals to measure large-scale and long-term temperature changes in the upper ocean layers [23–25]. The acoustic signal used in HIFT had a source level of 221 dB re 1 µPa at 1 m with a center frequency of 57 Hz [24], while the signal used in ATOC operated at 420 W (or 197 dB re 1 µPa at 1 m) and was centered at 75 Hz [26]. To address the concerns of potential impacts from the sound emitted, extensive field studies were conducted on marine mammals and other marine organisms, e.g., [27–32].

Besides industry and academia, the military—particularly, the naval community—produces intense underwater sound for various purposes. The sound sources include naval sonars, live-fire munitions, and underwater detonations used during training and testing activities and ship shock trials. However, the acoustic impacts from those sources were not broadly known until the early 1990s [33]. The situation changed dramatically in the late-1990s to early 2000s when several marine mammal mass-stranding events occurred in areas where the U.S. and North Atlantic Treaty Organization (NATO) navies had conducted exercises involving the use of active sonars, e.g., [34–38]. Those stranding events received considerable attention from environmental organizations, academia, and the public, which led to a surge in field and laboratory studies that have greatly increased our knowledge regarding the impacts of sound on marine mammals, as well as other marine life, including fish and invertebrates, in the past two decades [39–43].

Although our understanding of the impacts of sound on marine organisms has increased, the regulatory community has struggled to evaluate and incorporate new findings and data into impact assessments and environmental policies in general [44]. Since assessing acoustic impacts on marine life is an interdisciplinary field, it requires that regulators and policymakers have knowledge and education in both underwater acoustics and marine biology [45]. Therefore, a solid understanding of physical principles in acoustics is imperative for assessing the impacts of underwater sound.

This paper addresses many of the physical principles in underwater acoustics that have been and currently are applied to the regulation and management of underwater sound and what information needs to be obtained in the future.

## 2. Application of Underwater Acoustic Principles in Marine Conservation and Policy—Current Status

Impact assessments of various anthropogenic sound-generating activities involve the evaluation of the physical characteristics of the sound sources and the propagation of sound in the marine environment. Most of the concepts used in these assessments are based on the field of underwater acoustics. These include acoustic parameters, the characterization of underwater sources (measurements and modeling), the application of the sonar equation, and sound propagation modeling.

### 2.1. Acoustic Parameters

A number of physical quantities can be used to describe underwater sound in terms of acoustic energy (in joules), power (in watts), intensity (in watts per unit area), and pressure (in pascals or micropascals or μPa). However, more commonly, the quantity of sound is expressed in a relative unit of decibels (dB), which is a logarithm (base 10) ratio of a physical quantity to a reference quantity, as expressed in the following equation:

$$\text{SPL} = 10 \log_{10} \left( \frac{p}{p_0} \right)^2 \tag{1}$$

where SPL is the sound pressure level, $p$ is the acoustic pressure, and $p_0$ is the reference acoustic pressure.

This expression converts the physical unit to a "level". For example, the sound pressure (in μPa) can be expressed in the sound pressure level (SPL) in dB in reference to 1 μPa (dB re 1 μPa), which is the standard reference unit in underwater acoustics. For airborne sound, the standard referenced sound pressure level is 20 μPa. The mismatch of acoustic impedance between water and air due to the differences in the sound speeds of these two media results in different acoustic pressures from a source with the same acoustic intensity. Specifically, for a plane wave with a far-field intensity of $I$, the underwater acoustic pressure $p_w$ and airborne acoustic pressure $p_a$ are

$$\begin{aligned} p_w &= \sqrt{I \rho_w c_w} \\ p_a &= \sqrt{I \rho_a c_a} \end{aligned} \tag{2}$$

where $c_w$ and $c_a$ are the sound speed in water and air, and $\rho_w$ and $\rho_a$ are the density of water and air, respectively. The product $\rho c$ is referred to as a characteristic acoustic impedance. Given that the nominal sound speed in water is 1500 m/s, the nominal sound speed in air is 340 m/s, the density of water is about 1000 kg/m$^3$, and the density of air is about 1.225 kg/m$^3$, the underwater acoustic pressure from a sound source with the same intensity would be approximately 60 times great than that in the air.

For example, for a sound source with an intensity of 1 W/m$^2$, the underwater and airborne acoustic pressures would be approximately 1225 Pa and 20 Pa, respectively, based on Equation (2), and the underwater and airborne SPLs would be 182 dB re 1 μPa and 120 dB re 20 μPa, respectively. These issues often create confusion among lay persons and regulators who may not be well-versed in physical acoustics [46].

Notwithstanding the simple definition of SPL provided herein, several variations of broadband "sound levels" are tailored to address different types of source characteristics that are pertinent to various marine organisms that have different vibroacoustic sensitivities[1] and exhibit varying responses [47,48]. Some of the commonly used sound levels are the peak sound pressure level ($L_{\text{pk}}$, $L_{\text{0-pk}}$, or SPL$_{\text{pk}}$); root mean square (rms) sound pressure level ($L_{\text{p,rms}}$ or SPL$_{\text{rms}}$); sound exposure level ($L_E$ or SEL)l single-strike (single-shot or single-ping) sound exposure level ($L_{\text{E,ss}}$, $L_{\text{E,sp}}$, or SEL$_{\text{ss}}$); and cumulative sound exposure level ($L_{\text{E,cum}}$ or SEL$_{\text{cum}}$). The usage of these sound level metrics is summarized in Table 1.

---

[1] The authors acknowledge the importance of particle motion. However, particle motion, velocity, and acceleration are beyond the scope of this paper. Please see [47,48] for more details regarding particle motion.

**Table 1.** Summary of the sound level metrics commonly used in assessing the impacts of underwater sound on marine life.

| Metric & Notation | Equation for Derivation | Usage in Impact Assessment |
|---|---|---|
| Peak sound pressure level ($L_{pk}$, $L_{0\text{-}pk}$, or $SPL_{pk}$) | $L_{pk} = 10\log_{10}\left(\frac{p_{pk}}{p_0}\right)^2$ | The maximum instantaneous sound pressure, which is used to assess a potential permanent threshold shift (PTS) and temporary threshold shift (TTS) in the hearing of marine mammals [49–51], gastrointestinal tract injury in marine mammals [52], and mortality and injury in fish and sea turtles [53] exposed to impulsive sound. |
| Root-mean-square sound pressure level ($L_{p,rms}$ or $SPL_{rms}$) | $L_{p,rms} = 10\log_{10}\left[\frac{1}{T}\int_T \frac{p^2(t)}{p_0^2}\,\mathrm{d}t\right]$ | The square root of the average of the sound pressure squared over a given duration, which is used to assess potential behavioral disturbance in marine mammals [54] from impulsive and non-impulsive sound exposure—a time window that consists of 90% of the acoustic energy is used to calculate $L_{p,rms}$ for impulsive sound [55]. It also is used to assess the potential mortality, injury, or TTS in fish and sea turtles exposed to non-impulsive sound [53]. |
| Sound exposure level ($L_E$ or SEL) | $L_E = 10\log_{10}\left[\frac{1}{T_0}\int_{T_{100}} \frac{p^2(t)}{p_0^2}\,\mathrm{d}t\right]$ | A 1-s normalized $L_E$ is used to characterize the source level for non-impulsive sound [56]. |
| Single-strike, single-shot, single-ping sound exposure level ($L_{E,ss}$, $L_{E,sp}$, or $SEL_{ss}$) | $L_{E,ss} = 10\log_{10}\left[\int_{T_{100}} \frac{p^2(t)}{p_0^2}\,\mathrm{d}t\right]$ | For impulsive or non-impulsive intermittent sounds, this is the $L_E$ for a single hammer strike for pile driving [56,57], a single air gun shot for a seismic survey, or a single ping for sonar. |
| Cumulative sound exposure level ($L_{E,cum}$ or $SEL_{cum}$) | $L_{E,cum} = 10\log_{10}\left[\int_{T_{cum}} \frac{p^2(t)}{p_0^2}\,\mathrm{d}t\right]$ | This is the $L_E$ for the entire duration of sound exposure. It is used to assess potential PTS and TTS in marine mammals when exposed to impulsive or non-impulsive sounds [49–51] and the mortality or injury of fish and sea turtles exposed to impulsive sound [53]. |

Notation: $p_{pk}$ = peak acoustic pressure in a time series, $p(t)$ = time varying acoustic pressure in a waveform, $p_0$ = referenced acoustic pressure, which is 1 µPa, $T$ = duration of the time series, $T_{100}$ = the entire (100%) time duration of the time series, $T_0$ = a referenced time interval of 1 s, and $T_{cum}$ = the entire duration of sound exposure.

## 2.2. Source Characterization

In acoustics, a source is a physical device or object that generates acoustic disturbance(s) in a medium. A simple point source can be viewed as a pulsating sphere with its radius varying sinusoidally with time. The acoustic pressure generated by such a sphere is time-varying and contains one or more frequencies.

Similar to almost all real-world sources, very few anthropogenic sources can be treated as a simple point source. Sound sources that have routinely been evaluated for adverse impacts on marine mammals include seismic air guns, military sonars, various types of in-water pile driving, underwater detonations, drilling, and, to some extent, civilian sonars and high-resolution geophysical (HRG) devices. Although it is well-recognized that vessel noise is the most pervasive source of anthropogenic sound both in terms of temporal and spatial extents in the marine environment [58,59], its potential impacts are not well-addressed, nor is it currently regulated in most countries. Additionally, with the exception of certain military and civilian sonars, the majority of these sound sources are considered broadband.

Based on the temporal characteristics and the types of impacts[2], underwater sound sources are classified by the following categories: impulsive, non-impulsive, continuous,

---

[2]    In the U.S. regulatory framework, impacts on marine mammals are classified into two categories: Level A harassment, which has the potential to cause injury, and Level B harassment, which has the potential to cause behavioral disturbance, as well as temporary threshold shifts (TTS).

and intermittent. It should be noted that the definitions of these four categories within the regulatory community generally are qualitative, although quantitative methods have been proposed in a few cases when clear-cut distinctions between categories are evident. For example, when differentiating between impulsive and non-impulsive sources, a 3-dB difference in measurements between the continuous and impulse settings of a sound level meter (SLM) has been used [57]. Specifically, if the SLM measurement from the impulse setting (a 35-ms window) is 3 dB or greater than the continuous setting (a 1-s window), the sound should be classified as impulsive [60]. A recent study by Martin et al. [61] used the kurtosis of a 1-min time window to determine whether a sound was impulsive or non-impulsive.

However, not all of these categories are mutually exclusive. For example, a source that is impulsive is typically intermittent (e.g., impact pile driving), but a source that is non-impulsive can be either continuous (e.g., vibratory pile driving and removal) or intermittent (e.g., sonar). In addition, not all sources fit into a single category. For example, down-the-hole (DTH) pile installation produces both percussive hammering and continuous drilling sounds, while HRG devices can emit impulsive or non-impulsive intermittent sounds. Some common examples of the source categories used by the U.S. regulatory community are provided in Table 2. An explanation of how these different categories of sound sources should be analyzed under the MMPA is provided in a User Spreadsheet Tool by the National Marine Fisheries Service [62][3].

**Table 2.** Examples of common categories of sound sources regulated by the U.S. regulatory community.

| Source Type | For Assessing PTS and TTS | For Assessing Behavioral Disturbance |
|---|---|---|
| Seismic air gun | Impulsive | Intermittent |
| Impact pile driving | Impulsive | Intermittent |
| Underwater detonations | Impulsive | Intermittent |
| Vibratory pile driving and removal | Non-impulsive | Continuous |
| DTH pile installation | Impulsive and non-impulsive | Intermittent and continuous |
| Sonar | Non-impulsive | Intermittent |
| HRG devices | Non-impulsive and impulsive | Intermittent |
| Drilling | Non-impulsive | Continuous |
| Icebreaking | Non-impulsive | Continuous |

Notation: DTH = down-the-hole and HRG = high-resolution geophysical.

For the most part, source levels are based on broadband sound levels measured at given locations back-calculated to 1 m from the source or modeled (in which case, the spectra also are considered). For in-water pile driving for construction activities, the term "source level" used by the regulatory community in the United States typically refers to the broadband sound level ($L_{pk}$, $L_{p,rms}$, or $L_{E,ss}$) measured at or normalized to 10 m as opposed to the more conventional 1 m from the pile, e.g., [56]. For seismic air guns, source levels are obtained from in situ measurements at various distances back-calculated to 1 m from the source, e.g., [63–67]. For many sources for which measurements are not available, source models (e.g., Gundalf, Nucleus, and Airgun Array Source Model (AASM)) are used to estimate the source levels that then are fed into sound propagation models, e.g., [68].

[3] The User Spreadsheet Tool is available at https://media.fisheries.noaa.gov/2020-12/2020_BLANK_USER_SPREADSHEET_-508_DEC.xlsx?null (accessed on 2 February 2021).

### 2.3. Sound Propagation

As with all underwater acoustic analyses, the basic sonar equation with only the geometric spreading loss term is most commonly used by conservation biologists and regulators to estimate received sound levels. That equation is expressed in dB as

$$\text{TL} = F \log_{10}(R) \tag{3}$$

where TL is the transmission (or propagation)[4] loss [4,45,69–72], *F* is a coefficient for the TL, and *R* is the distance from the source to receiver (i.e., the animal). For a simple point source within a lossless infinite medium, *F* is 20, which implies the "spherical spreading" of acoustic energy. In a shallow-water environment, the boundary condition dictates that the acoustic energy predominantly follows a "cylindrical spreading" model, where the transmission loss would be expressed as $10\log(R)$ [4,69]. In addition, there is a "combined spreading loss" model that calculates transmission loss using spherical spreading to a certain range *H* where the sound reaches the sea floor—after which, cylindrical spreading is assumed [73].

The combined spreading loss is expressed as

$$\text{TL} = 20 \log_{10}(H) + 10 \log_{10}\left(\frac{R}{H}\right) \tag{4}$$

at a range (*R*) greater than the water depth (*H*). Although additional loss mechanisms such as absorption and scattering (i.e., volume and boundary scattering) also contribute to the decay of acoustic intensity over range, models that incorporate absorption and scattering terms are seldom used by regulators, mainly due to the fact that such models cannot be solved analytically. Similarly, transmission loss models that incorporate low-frequency cutoffs or leakages in shallow water also are rarely used by regulators.

To account for the additional losses due to absorption and scattering, and to partially account for acoustic energy that is confined within the boundaries, regulatory agencies often use 15 (i.e., the arithmetic mean between 20 and 10) as the transmission loss coefficient and define it as "practical spreading". The practical spreading model primarily is used to assess the impacts from pile-driving activities, e.g., [74]. Other transmission loss coefficients that have been used include the derivation of decay slopes from linear fit models of field measurements at varying distances, e.g., [75]. However, transmission loss coefficients obtained using field measurements are location- and season-specific, because received sound levels at distances from the source are products of multiple attenuation mechanisms. Factors such as sediment type, bathymetry, and temperature/salinity profiles of the water column often dictate far-field sound level measurements.

However, sophisticated numerical sound propagation models (such as ray theory, wavenumber integration, normal mode, the parabolic equation, etc.; see [69]) generally are not used by the regulatory community. Regulatory agencies typically rely on results provided by applicants or their contractors who have those modeling capabilities, e.g., [76,77]. In those cases, it sometimes is unclear whether the regulatory agencies adequately evaluated or validated the modeling results.

### 2.4. Impact Assessment Analyses

In general, the underlying approach for assessing the impacts of underwater sound on marine life uses a source–path–receiver model, where the source is the anthropogenic sound emitted, the path describes the assumed sound propagation, and the receiver is the animal(s) that detects the sound.

---

[4]　The authors recognize the difference between "transmission" and "propagation" under certain circumstances, where "transmission" could mean traveling of the acoustic wave from one medium to another. However, these two terms are used interchangeable herein, because the term "TL" is more widely used for "transmission loss" than "PL" for "propagation loss" in the underwater acoustics literature; see [4,45,70–73].

This model can be best presented in the form of a simple passive sonar equation:

$$RL = SL - TL \tag{5}$$

where RL[5] is echo level or received level, SL is source level, and TL is transmission loss [4,71].

The metrics used for received levels mirror those of acoustic parameters described herein; however, the numerical values of the levels, or thresholds, have been revised, as more studies have been conducted investigating the acoustic impacts of underwater sound. For example, the auditory injury thresholds (defined as permanent threshold shift (PTS)) were revised from the $L_{p,rms}$ thresholds of 180- and 190-dB re 1 μPa for cetaceans and pinnipeds, respectively [54], to the dual criteria of $L_{E,cum}$ and $L_{p,pk}$, with the incorporation of frequency-based, auditory weighting functions for the $L_{E,cum}$ metric [49,50].

The receivers that are pertinent to impact assessment analyses include all aquatic organisms that are sensitive to underwater sound and vibroacoustic disturbance. The levels upon which adverse impacts occur depend on the taxonomy, physiology, and behavioral ecology of specific species or individual animals, which is not within the scope of this paper. Interested readers are referred to several research, review, and guidance papers for the relevant information, e.g., [45,49–53,60,78].

The statistical detection theory at the receiver (i.e., the animal) is not currently considered in impact assessments of underwater sound. Such considerations would include quantitative studies of detection thresholds, the minimum signal-to-noise ratio needed to perceive the signal, the frequency spectrum and bandwidth of the signal, and the ambient sound, as well as receiver operating characteristic (ROC) curves, which describe the probability of detection at the receiver given a detection threshold and the signal-to-noise ratio [5].

Additionally, receiver (animal) movement modeling can be used to better inform an impact assessment by estimating the number of animals, in the form of "animats" that could be affected (taken). Animal movement modeling falls within the field of behavioral ecology; therefore, it is not discussed further. However, multiple animal movement models do exist; see [79] for information on the Marine Mammal Movement and Behavior (3MB)[6] and [80] for information on the Navy Acoustic Effects Model (NAEMO).

### 2.5. Chronic Impact Assessment and Soundscape Analyses

Over the past decade or so, there has been increased interest in addressing potential chronic and cumulative impacts from low-intensity sound sources (e.g., commercial ships and smaller vessels) that are not typically regulated [81,82]. Many of these studies have shown that chronic exposure to low-intensity sound can cause various adverse effects, such as communication masking, changes in vocalizations and echolocation, and increased stress levels [83].

With the recent advances in underwater acoustic sensing technology available to nonmilitary researchers, the accessibility of large acoustic datasets from global sensor networks, and the enhanced computational resources for signal processing of large acoustic datasets, the large-scale, long-term monitoring of the underwater acoustic environment is feasible. These new opportunities have created considerable possibilities for studying the relationship between underwater acoustic and biological phenomena [84].

Many of these studies build on earlier research on ambient sound by analyzing spectral contents of long-term acoustic recordings. A frequency–time analysis has been used to investigate the inter-relationships of three sound types—biophony, geophony, and anthrophony—within an ecosystem. This relatively new subfield, ecoacoustics and

---

5   In most underwater acoustics literature, "EL" (echo level) is typically used to indicate the received (echo) level at the receiving transducer, and "RL" is reserved for the reverberation level in the sonar equation (e.g., [5,60,70,73]. However, this paper uses "RL" to indicate "received level", which is a more common practice within the ocean sound community (e.g., [39]).

6   3MB is available at http://oalib.hlsresearch.com/Sound%20and%20Marine%20Mammals/3MB%20HTML.htm (accessed on 8 February 2021).

soundscape ecology [85,86], takes a holistic approach for studying underwater sound and their relationship to marine life. Ecoacoustics and soundscape ecology allow for the assessment of the overall health of the ecosystem by including the acoustic component, a very important element that has been long overlooked.

### 2.6. Knowledge and Expertise of Regulatory Community

The regulatory community that oversees the implementation of marine conservation and policy measures concerning the impacts of underwater sound primarily are composed of conservation biologists and environmental policy specialists, many of whom lack a formal educational background in physics, mathematics, or underwater acoustics. Staff analysts and managers who conduct impact assessments and make regulatory decisions may receive on-the-job training through seminars and web-based tutorials, such as those on the Discovery of Sound in the Sea website (https://dosits.org/, accessed on 2 February 2021). However, such ad hoc training is inadequate to bring analysts within the U.S. regulatory community beyond the level of performing simple analytical calculations of sound propagation using scripted spreadsheets. Few are able to evaluate sophisticated acoustic models or sound source measurements. The regulatory agencies have yet to prioritize the knowledge and skills of physical acoustics that are necessary to conduct impact assessments of underwater sound. These shortcomings have resulted in frequent errors, the omission of pertinent information, and inconsistencies in agency decision-making documents, as documented in multiple comment letters from the U.S. Marine Mammal Commission, an independent oversight agency, e.g., [87–96].

For example, when addressing the potential impacts of the relatively novel DTH pile installation method, one regulatory agency repeatedly mischaracterized the source, used inappropriate thresholds, and underestimated the source levels, which resulted in much smaller impact zones [89,92–94]. In another example, the same agency fabricated a method termed "log average of the sources"—taking a log average of log-based sound levels to derive a source level for DTH pile installation—which is not rooted in the principles of underwater acoustics [89]. The agencies also have routinely used inappropriate and inconsistent source levels for pile driving and removal, as well as inappropriate thresholds in general [87,88] and inappropriate assumptions and inputs for estimating the extents of the various impact zones [89,91–93]. The aforementioned issues result in inaccurate and often underestimated impact zones, which are used to determine whether and how an animal may be affected and to inform the mitigation measures necessary to minimize those impacts.

## 3. Needs for Using Underwater Acoustics in Marine Conservation

While underwater acoustic concepts are well-understood, information gaps exist, and training is necessary to improve the accuracy of and consistency among impact assessments. Data and information needs include source characteristics of novel sound sources; robust and easy-to-use sound propagation models; and statistical detection models, as well as quantitative exposure models that evaluate the acute, chronic, and cumulative impacts on marine organisms. While the last topic is addressed primarily by the field of bioacoustics, the knowledge of behavioral ecology and physiology needs to be incorporated into any impact assessment [38] and is beyond the scope of this paper. The research needs regarding source characterization, propagation modeling, and detection theory are provided herein.

### 3.1. Source Characterization

Despite the numerous technical reports that involve measurements of underwater sound generated by various sound sources, robust data are still lacking concerning some of the sound sources. Those deficiencies include: (1) novel sound sources, (2) uncommon sources for which few data exist, and (3) a lack of scientific rigor in measurements. Large variations in source levels also are evident among the same source type, which adds another complicating factor.

Some sound sources are novel, which either have not been used in the marine environment until recently or have not been well-documented. For example, in-water pile installation using DTH pile installation is a relatively new application in the marine environment and uses a combination of percussive and drilling mechanisms [57]. During DTH pile installation, a percussive hammer acts directly upon the bedrock to create a hole for the pile to enter, while the drill cuttings and debris at the rock surface are removed by an airlift exhaust through the inside of a pile. Therefore, the sound generated from DTH pile installation contains both impulsive, intermittent components (from percussive hammer strikes) and non-impulsive, continuous components (from drilling actions and airlifts of debris). Currently, only a few studies have conducted measurements of DTH pile installations [57,97–101]. Additionally, in situ measurements of DTH pile installations have been limited to piles with diameters of only 0.20 m, 0.46 m, 0.61 m, and 1.07 m. Those data are scant and inadequate, particularly for larger-sized piles. Piles used in coastal construction projects can be larger and are generally much larger for offshore wind turbine structures. In addition, the substrates associated with the measured sound levels often are not specified.

Other sound sources that lack the full complement of the relevant acoustic information include various nonmilitary shipboard or towed sonars, transducers, other HRG sources, and acoustic deterrent devices. While potential effects from exposure to these sources are still under debate due to their generally lower intensity and high-frequency components (which are subject to greater absorption losses) [102–105], the source levels of these devices have not been well-documented beyond the manufacturer specifications, e.g., [106].

There also are new sources that are being developed. One example is marine vibroseis, a source that is being developed to replace a conventional air gun array with much reduced SPLs [107–109]. Source characteristics of marine vibroseis are mostly modeled; there are few in situ measurements of sound levels of marine vibroseis to date [110].

While a number of models were developed and countless measurements were made for open-water underwater detonations years ago (as one of the sound sources for underwater acoustics research was small charges), e.g., [4,111–114], few studies are available on sound characteristics and propagation from detonations that are embedded in bedrock or other structures, e.g., [115,116]. The lack of measurements from confined underwater detonations presents significant challenges when assessing environmental impacts for projects that use such methods, particularly for shipping channel deepening or structure removal.

Conversely, sound generated from marine seismic surveys using air guns have been well-documented since the 1980s [45]. Numerous measurements of seismic air gun arrays have been acquired in the Arctic for the purpose of environmental compliance from the mid-2000s to early 2010s, e.g., [63–67,117–119]. Nevertheless, due to differences in the volumes of air gun arrays and their deployment configurations, those measurements were only pertinent to those specific surveys. Industrial standard models such as Gundalf [120,121], Nucleus [122], and AASM [123] have been used to predict air gun array sound levels to form the sound propagation modeling used in impact assessments; however, none of the models are available to the regulatory community for use at this time. In addition, the accuracy of these models is still being evaluated by the underwater acoustic modeling community, e.g., [124,125].

Similar issues exist for in-water pile-driving data. Despite the fact that large quantities of pile-driving sound level measurements exist, e.g., [56], the regulatory community has struggled to use representative source levels consistently for specific pile materials and dimensions. Some of the inconsistencies are due to differences in bathymetry, substrate type, hammer energy, and other environmental parameters at the locations where measurements have been collected. An attempt is underway by one of the authors to review and analyze all available pile-driving measurement data and to recommend a set of "generic" 10-m normalized source levels based on the various pile types and diameters. Similar to air gun source models, models for pile-driving sound sources also exist, e.g., [126–128]. None

of these models are readily available for use by the regulatory community to form its environmental impact assessments.

In addition to the need to characterize broadband levels from many of the afore-mentioned sound sources, spectral information regarding these sources is in demand, as impact assessments of auditory effects are often frequency-dependent, especially for marine mammals [50].

Finally, most sound level measurements have been conducted for the purpose of environmental compliance and were collected and analyzed by contractors of regulated entities with varying professional experience and/or knowledge in underwater acoustics. Most of these measurements exist in the form of gray literature, e.g., [56], and few of them have been peer-reviewed or published in peer-reviewed journals. Therefore, the quality of some sound source measurements is questionable and should be evaluated further.

### 3.2. Sound Propagation Models

Although numerous models exist for underwater sound propagation [69], the majority of these models assume that the source is in the open water. However, sources from some of the regulated activities occur within sediment or structures (e.g., confined underwater detonations) or are coupled with the sediment (e.g., DTH pile installation). The authors are unaware of an available propagation model for these sources, despite the increasing use of these sources in recent years.

For the majority of sources that are used in a water column, the existing propagation model commonly used by the regulatory community is a simple spreading model with a transmission loss coefficient of 20 or 15, depending on the source. Given that underwater sound propagation is almost always a complex process that involves bathymetry and topography of the location, substrate layers and types, temperature and salinity profiles of the water column, sea surface conditions, and the frequency spectrum of the source, sophisticated numerical modeling typically is required to obtain the reasonably accurate results needed for impact assessments.

Although many of the sophisticated numerical propagation models are derived from well-established propagation theories [129], the implementation of these models is beyond the expertise of the regulatory community due to the lack of necessary technical skills within the agencies. For example, high-level programming languages such as MATLAB or Octave are not among the standard software used by the U.S. regulatory community for conducting impact assessments.

Given these resource and technical limitations, it is beneficial to develop relatively simple numerical models that can be incorporated into an Excel spreadsheet format. For example, the U.S. National Marine Fisheries Service has developed a simple spreadsheet tool that incorporates the frequency of absorption and beam width for determining sound propagation and estimating the distances at which behavioral harassment could occur in marine mammals [130].

In another example, a damped cylindrical spreading (DCS) model-based spreadsheet, or DCSiE, was recently developed with funding from the BOEM to estimate the distances of certain received sound levels from impact pile driving for offshore wind turbine installations [131]. This spreadsheet tool incorporates information related to bathymetry and the substrate type, in addition to the measured sound level at a reference distance (typically, no less than three times the water depth at the source). It is based on a reasonably simple but more accurate DCS model [132–134]. To implement DCSiE properly for estimating impact zones, one must have an understanding of the sediment composition and the layering of those sediments (including the sediment porosity and particle size) in the project area. Unfortunately, these data are lacking in most regions and are not routinely described when pile-driving measurements are collected. In addition, a comparable sound propagation model for vibratory pile driving and removal and DTH pile installation currently does not exist.

Although these spreadsheet tools can perform simple propagation modeling to a certain degree, they cannot replace sophisticated numerical models that are commonly used by the underwater acoustics community. To address such deficiencies, a standalone software package that does not require programming skills needs to be developed for the regulatory community.

### 3.3. Statistical Detection Models

None of the impact assessment tools for underwater sound currently address statistical signal detection at the receiver. The received level at the animal is therefore considered the level of exposure. Such an approach generally is acceptable when assessing the PTS or temporary threshold shift (TTS), as most data on those effects are based on direct measurements at the animals. However, they may not be accurate for quantitatively addressing behavioral disturbances and acoustic masking. Most research on marine animal audiograms and hearing thresholds is conducted in the absence of background sound or at very low ambient conditions with a higher signal-to-noise ratio than is typical in the marine environment. Only a few studies have addressed signal detection in the presence of noise, which could elucidate detection thresholds of some marine mammal species, e.g., [135–137] and review [138]. The authors are not aware of any such studies in species other than marine mammals. Although the detection theory falls within the fields of auditory physiology and behavioral psychology, the information from such studies is critical in the application of underwater acoustics to impact the assessments of sound. The lack of information on the auditory detection thresholds under various noise conditions by many marine species makes it impossible to conduct assessments of masking using the well-established statistical detection theory with ROC curves [5].

### 3.4. Needs for Chronic Impact Assessment and Soundscape Analyses

Despite the recent progress in understanding the impacts of chronic low-intensity sound on marine life (e.g., [83]), most of the regulatory community has been slow to implement the associated analyses. For example, shipping noises receive relatively little consideration during conservation planning and regulatory management. One of the reasons appears to be that the adverse effects from the low-intensity sound are difficult to quantify on a project and area basis, which is the main mechanism underpinning the various regulations. Therefore, models that can quantify fine-scale and project-specific impacts from low-intensity sound exposure should be developed. These models would be able to assess the energetic cost to marine life from sound exposure in the form of behavioral modification, changes in vocalizations and echolocation, communication masking, habitat displacement, and increased stress levels.

Another "low-level" impact that has received little consideration is reverberation—specifically, the reverberation field between intense intermittent sounds due to multipath propagation [139–141]. It has been suggested that the elevated background sound levels from reverberation have the potential to mask vital marine mammal acoustic cues [142]. However, there are few studies that provide quantitative data on the threshold level associated with auditory masking [143].

In addition, further studies are needed to investigate how the soundscape changes as a result of long- and short-term habitat modifications, which may affect certain species and, in turn, set off a cascade through various trophic levels and affect the ecosystem as a whole. While most of the questions being addressed lie in the field of ecoacoustics, the technical capability required to analyze large amounts of acoustic data that are being collected continuously by many global observation networks is a critical need to be addressed [144–146].

### 3.5. Needs for Expertise and Knowledge within the Regulatory Community

Last, but not least, advancements in the knowledge of underwater acoustics and its applications are not possible without a regulatory community that is well-versed in

underwater acoustics. Since assessing acoustic impacts on marine life is an interdisciplinary field that involves physical acoustics, oceanography, and biology, scientifically sound environmental policy and conservation measures can only be developed through a solid understanding of the scientific principles within these fields. Specifically, the knowledge, skills, and expertise in performing and evaluating numerical source and propagation models; acoustic measurements; and exposure and impact models are key areas where gaps currently exist. It is imperative for regulatory agencies to integrate professional physical acousticians into their hierarchy, rather than relying on policy experts that lack formal education in and an understanding of physics, mathematics, or engineering. As the renowned acoustician Dr. Allan D. Pierce stated, "a deep understanding of acoustical principles is not acquired by superficial efforts" [147].

## 4. Conclusions

The field of the environmental conservation that addresses the impacts of underwater sound on marine life has advanced considerably over the past half-century. Although one of the initial concerns involved ever-increasing ocean ambient impacts on the communication space of baleen whales [11], the field advanced most readily when acute impacts from ATOC sources, seismic air guns, and military sonar were investigated [17,27,34]. The environmental impact assessments of these sound sources were assisted largely by knowledge within the field of underwater acoustics, e.g., [29,39,45].

Over the years, assessing the impacts of underwater sound has gradually evolved into a research area with its own unique definition of acoustic parameters, sound sources, propagation modeling, and measures of biological impacts, e.g., [51,52,60]. Besides addressing the direct and acute impacts of sound, which is largely under the purview of natural resource agencies that implement various regulatory statutes and measures [49–52,54], recent developments in this field include research that addresses the overall acoustic environment. These new studies have broadened the scope of the relatively narrow-focused field that only addressed acute impacts into the emergent subfield of ecoacoustics and soundscape ecology [85,86]. Holistic approaches for studying underwater sound in relation to marine life allow for the assessment of the overall ecosystem health, which includes an acoustic component, and of certain elements that have long-term and chronic adverse effects on marine life (e.g., low-intensity but pervasive sound from ships).

As with all emerging scientific fields, many information gaps still exist and likely will exist for the foreseeable future. Sound characteristics of many known and novel emerging anthropogenic sound sources have yet to be assessed and validated. Data on sediment composition and associated layering are lacking in many regions, which compromises the integrity and accuracy of any sophisticated sound propagation model. Sound propagation modeling, though firmly established within the underwater acoustics field, needs to be made accessible to the regulatory community, which is largely composed of conservation biologists and environmental policy specialists. Simple analytical computational methods and/or standalone software that does not require programming skills are desirable and need to be developed for regulators to conduct impact assessments. Finally, prioritizing the hiring of scientists who have formal educations in physics, mathematics, or engineering to co-lead or co-manage environmental impact assessments is essential to forming scientifically sound policy and conservation measures that minimize the impacts of underwater sound on marine life.

**Author Contributions:** Conceptualization: S.G. and T.B.; writing—original draft: S.G. and T.B.; and review and revisions: S.G., T.B., and J.V. All authors have read and agreed to the published version of the manuscript.

**Funding:** This research received no external funding.

**Institutional Review Board Statement:** Not applicable.

**Informed Consent Statement:** Not applicable.

**Acknowledgments:** The authors thank Peter Thomas for a thorough review of and comments on this manuscript. The authors also appreciate the constructive comments from the three anonymous reviewers.

**Conflicts of Interest:** The authors declare no conflict of interest.

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
