# Peer review of "Use of Underwater Acoustics in Marine Conservation and Policy: Previous Advances, Current Status, and Future Needs"

_jmse, doi:10.3390/jmse9020173_

Round 1

Reviewer 1 Report

Underwater anthropogenic noise and its effects on marine life has received special attention from the scientific community during the last two decades. Such an active field deserves periodic updates of the state of the art and review papers are welcome. The paper entitled "Use of Underwater Acoustics in Marine Conservation and Policy: Previous Advances, Current Status, and Future Needs" is a good attempt to introduce not specialized readers in the field, placing it in the broader framework of underwater acoustics, and to make an account for the present status of the application of underwater acoustics for marine conservation and policies, of scientific trends and remaining needs for that purpose. To my understanding the paper is very well structured and it is easy to be read despite the big amount of provided information and references. Nevertheless, it could be improved to offer a more complete picture of the knowledge gaps, and the actual need of technological and management tools. My main observations/objections are the following:

- the exposed motivation is mainly centered in the long-studied effects on marine mammals and the attention paid to 1990s-2000s stranding episodes. Being truth the triggering effect of those studies and events, at international level it must be referenced the increasing social awareness of the degradation of oceans that led to establish protection policies like the Marine Strategy Framework Directive (MFSD), adopted by the European Union in 2008 to protect more effectively the marine environment. Underwater noise and its effect on ecosystems was set as one of 11 descriptors of good environmental status of oceans in the frame of the MFSD and research in the field was boosted with unprecedented intensity.

- in close relationship with the wider scope of the expected effects of underwater noise on marine life, references 37-40 account for such effects on invertebrates and fishes, but no description is given in the submitted paper. Even more, in its following sections most of the citations about effects caused by different noise sources are exemplified with mammal studies which are probably closer to the specialization and interest of the authors, but provide a partial picture. I would suggest to complete such descriptions with references to other organisms. This consideration should be taken into account along sections 2 and 3.

- connected with this involuntary partial view, I find a fundamental gap in the description of the acoustical parameters of subsection 2.1: while mammal hearing is based on detection of sound pressure, fish and invertebrates (i.e. most aquatic animals) primarily sense sound using particle motion. Some references can be found in https://doi.org/10.1121/1.3596464 and https://doi.org/10.1111/2041-210X.12544, by instance.

- these last two observations are directly connected with the description of needs for using acoustics in marine conservation. Hydrophone based systems are commercially available, but equipment and processing software for particle-motion measurements are not so easily accessible. This can also be stated about calibration procedures. Still proper underwater low frequency calibration procedures of hydrophones and complete electro-acoustic chains are under consideration and particle-motion calibration standards are also needed (http://eprintspublications.npl.co.uk/id/eprint/7088). Consequently, and despite the on-going works, there is a lack of research about the role of particle displacement in the suggested soundscape ecology.

- Subsections 2.2 and 3.2 are devoted to Sources characterisation. The characterisation of ship as a source is not mentioned, being fundamental knowledge for shipping noise modelling and the discussion for status and needs of subsections 2.3 and 3.2. Again there is a certain “bias” in the paper, more addressed to mammal protection policy than a more general consideration of marine ecosystems.

I would ask to the authors to include all these considerations and revise the concept of the paper, or to limit its scope to the application of acoustics to marine mammal protection policy.

Author Response

Please see attached response to reviewer comments

Reviewer 2 Report

In general the manuscript is overly detailed in some areas and too superficial in other areas. It makes a good point that regulators are basically at the mercy of permit applicants because the latter have much more extensive understanding of underlying acoustics and also in the current regulatory environment many critical factors are ignored. Of course they are ignored because they are exceedingly complex. This manuscript could reasonably be address is aware of these problems so its contribution to the overall literature is minimal.

Specific points:

Line 34: Schevill and Lawrence 1949 predate the Kellogg et al reference by 4 years and should be cited instead.

Line 34: Ray et al 1969 was the first positive evidence of song in a marine mammal.

Line 48: While the tuna/dolphin issue was instrumental in the passage of the MMPA, fisheries received major exemptions from many of the regulations. They are exempt from most incidental take regulations. This should be clarified. 

Lines 102-113: This is too simplified to be useful and will create more confusion than enlightenment. There are much better explanations of the difference in dB between air and water--e.g., https://fas.org/man/dod-101/sys/ship/acoustics.htm. The bottom line is that when taking into consideration both the reference level and the acoustic impedance differences the difference is 62 dB. Further what is important in such jumbo jet and military sonar comparisons is the loudness level (i.e., dB above threshold) for a human hearing a jumbo jet and a cetacean hearing a military sonar based on their respective frequency weighting curves. Substantial revision is required.

Line 123 - Table 1: Why add irrelevant information? They should stick to the metrics used in marine mammal studies. 

Line 123 - Table 1: Single Strike, Single Shot...what is [52,53]0)? I know the references, the rest appears to be a typo.

Line 163 - Table 2: Left alignment within table cells will read better.

Line 214: Apparently the common use of RL instead of EL is a pet peeve of one of the authors, but as noted RL is the common expression so use it in the text and confine the pet peeve to a footnote.

Lines 284-287: This paragraph is contains several non sequiturs. The majority of sound sources are novel? The majority of sound sources are uncommon? 

Lines 404-405: Citations are needed in reference to "the recent progress in understanding impacts of chronic low-intensity sound on marine life." Identify the recent progress.

Line 447: It should be noted here that almost all the relevant regulatory data on biological impact came from studies on captive marine mammals.

Author Response

Please see the attached response to reviewer comments.

Reviewer 3 Report

This paper outlines the principles of underwater acoustics, standard metrics for quantifying the sound levels and how sound is applied in environmental impacts assessments including gaps in those efforts. Finally, the authors present several recommendations for improvements to the impact assessment process and regulatory community. While these recommendations are sound and have merit for the community to be aware of, JMSE seems an ill-fitting journal for publication. As there was no (experimental or theoretical) study related to marine science and engineering presented in the paper, the work is out of scope for JMSE. This decision is up to the Editor. However, I would like to note the content is far better aligned with Marine Science and its readers.

The paper makes several strong points against the current practices of the regulatory community, namely use of overly simplified propagation models and lack of personnel knowledge on the subject. It would be very helpful to provide specific examples and resulting consequences of these issues. There is one mention of shortcomings documented by the MMC but no details are given within the text of the paper. To ensure credibility to the statements made, the MMC findings need to be transparent. 

Author Response

Please see the response to reviewer comments.

Round 2

Reviewer 2 Report

This is a significant improvement over the initial submission.

In reading the addition in lines 314-326 the thought occurred to me they might want to consider as well Tyack and Thomas 2019 "Using dose–response functions to improve calculations of the impact of anthropogenic noise" which clearly demonstrates how using a criterion threshold and assuming all animals within range of that threshold are affected and none beyond significantly underestimates the number impacted by the source. This is also relevant to the text on lines 435-439.

Reviewer 3 Report

No further comments.